# Stall Torque Performance Analysis of a YASA Axial Flux Permanent Magnet Synchronous Machine

**Jordi Van Damme** [1,2,*], **Hendrik Vansompel** [1,2] and **Guillaume Crevecoeur** [1,2]

1    Department of Electromechanical, Systems and Metal Engineering, Ghent University, 9000 Ghent, Belgium; hendrik.vansompel@ugent.be (H.V.); guillaume.crevecoeur@ugent.be (G.C.)

2    Flanders Make@UGent-MIRO, 9052 Ghent, Belgium

*    Correspondence: jordi.vandamme@ugent.be

**Abstract:** There is a trend to go towards low gear-ratio or even direct-drive actuators in novel robotic applications in which high-torque density electric motors are required. The Yokeless and Segmented Armature Axial Flux Permanent Magnet Synchronous Machine is therefore considered in this work. In these applications, the motors should be capable to deliver high torque at standstill for long periods of time. This can cause overheating of the motors due to a concentration of the losses in a single phase; hence, it becomes necessary to derate the motor torque. In this work the influence of the slot/pole combination, the addition of a thermal end-winding interconnection and the equivalent thermal conductivity of the winding body on the torque performance at standstill will be studied both experimentally via temperature measurements on a prototype stator, and via a calibrated 3D thermal Finite Element model. It was found that both a good choice of the slot/pole combination and the addition of a thermal end-winding interconnection have a significant influence on the torque performance at standstill, and allow up to 8% increase in torque at standstill in comparison to a reference design.

**Keywords:** axial flux; end winding; slot/pole combination; stall torque





## 1. Introduction

Owing to its high torque density and energy efficiency, Yokeless and Segmented Armature (YASA) Axial Flux Permanent Magnet Synchronous Machines (AFPMSM) have proven their benefits in several application areas such as transportation and wind energy generation [1,2]. This axial flux motor topology consists of two rotors and a central stator, as illustrated in Figure 1. Motivated by the merits in these applications, more recently, this machine topology was also considered for use in quasi-direct-drive and direct-drive actuators of novel robotic applications [3,4].

In some robotic applications however, the electric motor has to generate its maximum torque at (quasi-)standstill during a significant fraction of the load cycle. This is, e.g., the case in force-controlled robotic grippers [5] (See also Figure 1), or when a robot has to hold a vertical load against gravity for a long time [6]. This leads to a concentration of the losses in a single phase of the motor. For the worst-case commutation position: twice the rated conduction losses are dissipated in a single phase when the motor produces its rated torque at standstill. Although the sum of the conduction losses of all phases in this case does not exceed the rated conduction losses, due to the uneven loss distribution, this can eventually lead to overheating of the motor at standstill. This problem is well-known in academic and industrial literature [6,7]. To avoid overheating, the stall torque, i.e. the maximum torque that a motor can produce under certain cooling conditions in steady-state at standstill, is typically lower than the rated torque.

Alternatively, brakes can be used to generate a stall torque and hold a load. There exist various kind of brakes [8], and most of them use an electromagnetic actuator such as

a solenoid to engage or disengage friction discs. These friction discs provide the required holding torque without consuming energy. Hence, they can generate a higher stall torque. However, in certain applications, e.g., force-controlled robotic grippers where soft objects need to be grasped, the force of the gripper needs to be precisely controlled in order to avoid damage to the soft object [5]. Traditional grippers that use brakes cannot be used in this case since they rely on applying a sufficiently large torque to hold the object and then engage their holding brake. This would harm the soft object. To conclude, brakes can generate much higher stall torques; however, they are not usable in every application.

So far, no solutions have been presented yet to overcome the problem of overheating at standstill. Researchers have predominantly focussed on strategies that improve the overall thermal performance of YASA AFPMSMs through either an improved cooling (i.e., liquid instead of air cooling [9,10]), through impregnating the windings [2] or through the introduction of radially inward heat extraction fins [1]. Although these strategies are very effective in increasing the total dissipated power losses, they are less effective in mitigating local overheating due to the uneven loss distribution caused by high torque at (quasi-)standstill. The problem of uneven temperature distribution due to partial immersion of the stator in an oil-cooled outer rotor radial flux machine was discussed in [11]. Oil storage slots were presented as an effective solution to reduce the local overheating. Although the presented solution appears to be very effective at rated speed, at (quasi-)standstill, the oil flows under the action of gravity to the bottom half of the motor, which drastically reduces its cooling effectiveness.

In this work, the problem of overheating during standstill due to the uneven loss distribution will be analysed. Based on physical insights from previous studies on the thermal behaviour of YASA AFPMSMs [1,2,9], three key design parameters have been identified that potentially have a significant impact on the redistribution of the heat from the phase with the highest losses to the phases with lower losses. The influence of these three key design parameters on the stall torque performance is studied in this work (See Figure 1):

1. *Slot/pole combination:* This aspect influences both the fundamental winding factor and back-emf constant, which directly impact the torque. It also influences the number of adjacent slots belonging to the same phase. As heat has to flow from the phase with the highest losses to phases with lower losses, it can be expected that the number of adjacent slots influences the thermal performance under uneven loss distribution.

2. *Thermal end-winding interconnection:* From previous studies, it is known that the end-winding at the inner diameter is often the hottest area of a YASA AFPMSM [2,10]; therefore, a good thermally conducting ring which interconnects all end-windings can redistribute the heat from the phase with the highest losses to the other phases.

3. *Equivalent winding body thermal conductivity:* Since in a YASA AFPMSM there is no iron stator yoke which has a good thermal connection with all slots, the equivalent thermal conductivity of the winding body can have a significant influence on the heat transfer between phases.

Note that also the axial flux machine topology can have an influence on the results, especially in the double stator, single rotor or single stator, single rotor topologies where the stator has an iron yoke providing a good thermal connection between the coils. However, this analysis falls beyond the scope of this work. The focus in this work is on the single stator, double rotor topology, which is known to have a higher torque density and energy efficiency [1,2].

The analysis results in this work will lead to a better understanding of the stall torque performance and can be used in the design of motors for applications where high torque at (quasi)-standstill is important. Other applications might also benefit from the results. For example, in [12,13], the problem of local overheating due to unbalanced supply voltage or inter-turn short circuits in induction motors was analysed. The studied key design parameters in this work can also improve the performance in these applications by redistributing the heat from the location with a higher loss concentration. However, this is beyond the scope of this article.

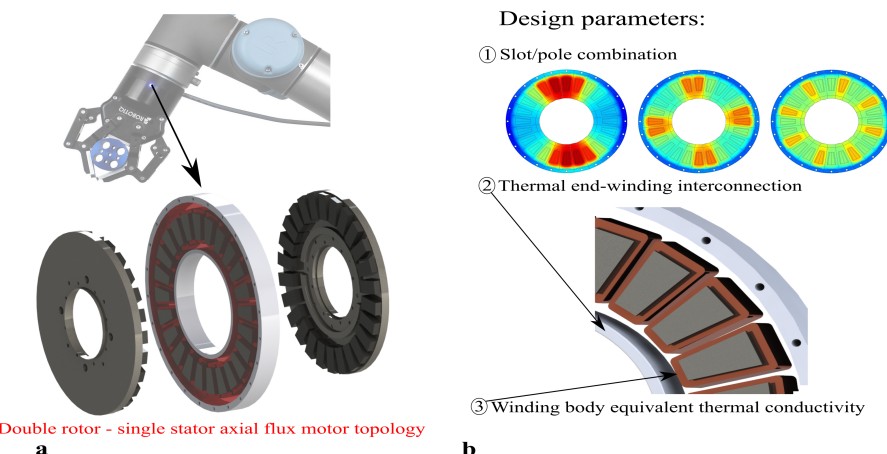

Design parameters:

① Slot/pole combination

② Thermal end-winding interconnection

③ Winding body equivalent thermal conductivity

Double rotor - single stator axial flux motor topology

**a**                           **b**

**Figure 1.** (**a**) The motor in a force controlled gripper has to generate high torque at (quasi)-standstill [14]. (**b**) Overview of the studied parameters in this work that potentially have a significant impact on the stall torque performance.

To study the influence of the aforementioned design parameters on the stall torque performance, a prototype YASA AFPMSM shown in Figure 2 is considered. Its main specifications are given in Table 1. The influence of the key design parameters will first be studied experimentally, the corresponding measurements will also be used to identify a 3D thermal Finite Element (FE) model. This model will allow to further analyse and improve the understanding of the experimental findings.

This paper is organised as follows: first, the three studied parameters are described in detail and illustrated for the prototype YASA AFPMSM in Section 2. Subsequently, the experimental setup is described in Section 3.1. The 3D thermal FE model is outlined in Section 3.2. Finally, the results of the experimental and simulation parameter study are presented in Section 4.

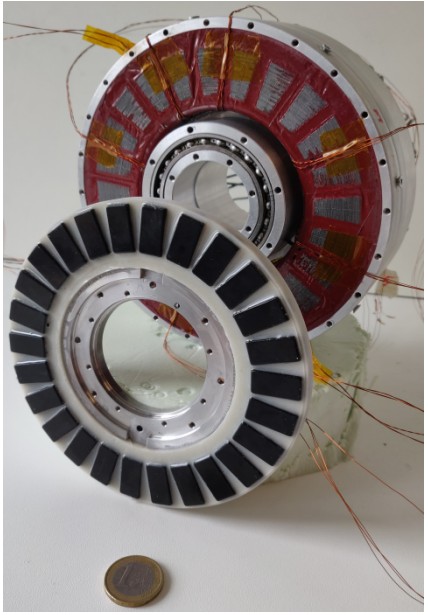

**Figure 2.** Prototype Yokeless and Segmented Armature Axial Flux Permanent Magnet Synchronous Machine used throughout this work; it has two rotors and a single stator. For visualization purposes only a single rotor is shown here. Its specifications are given in Table 1.

**Table 1.** Specifications of the test case YASA AFPMSM.

| Parameter | Symbol | Value | Unit |
|---|---|---|---|
| Three-phase inverter DC bus voltage | $V_{DC}$ | 48 | V |
| Maximum speed | $\Omega_{max}$ | 300 | rpm |
| Number of pole pairs | $N_p$ | 13 | / |
| Number of slots | $Q_s$ | 24 | / |
| Number of phases | $n_{ph}$ | 3 | / |
| Number of turns per tooth coil | $n_{turns}$ | 35 | / |
| Outer diameter stator iron core | $D_o$ | 138.5 | mm |
| Inner diameter stator iron core | $D_i$ | 98.5 | mm |
| Axial length stator iron core | $h_{stat}$ | 15 | mm |
| Axial slot length | $h_{slot}$ | 10 | mm |
| Total axial length (incl. housing) | $l_{tot}$ | 62.5 | mm |
| Slot width | $b_{slot}$ | 6 | mm |
| Airgap thickness | $h_{air}$ | 1.5 | mm |
| Magnet height | $h_{mag}$ | 5 | mm |
| Rotor yoke height | $h_{mag}$ | 6 | mm |

## 2. Design Parameters Affecting Tangential Heat Transfer

### 2.1. Loss Distribution

From previous work on thermal analysis of YASA AFPMSMs, it is known that there is no heat transfer between adjacent tooth coils under a uniform loss distribution and cooling [2]. However, when the motor produces torque at (quasi-)standstill, the phase currents are (quasi-)DC currents whose magnitude depends on the rotor position. Since the actual standstill rotor position is very application-specific and can vary with the operating scenario, the design should account for the worst-case standstill rotor position. This is the position where $I_u = 2 \cdot I_v = 2 \cdot I_w = \sqrt{2} \cdot I_{nom,RMS}$ [6]. With $I_u, I_v, I_w$ the (quasi-)DC phase currents and $I_{nom,RMS}$ the rated motor RMS current. Recall that for this worst-case position, the conduction losses in phase U equal twice the rated conduction losses and that the total conduction losses in the windings equal the rated conduction losses. A heat transfer between phases can be expected in this case, i.e., a heat transfer in the tangential direction. This worst-case scenario will be considered throughout this work. Note that phase *U* was chosen arbitrarily as the phase with the highest losses in this work.

### 2.2. Slot/Pole Combination

The slot/pole combination influences the stall torque performance in different ways. The (stall) torque is given by [15]:

$$T = 3 \cdot \xi \cdot k'_\phi \cdot I, \tag{1}$$

with $\xi$, the fundamental winding factor, $k'_\phi$, the back-emf constant and $I$ the RMS phase current. A first way is via the fundamental winding factor. It is given by the product of the pitch factor $\xi_p$ and distribution factor $\xi_d$. For a 3-phase, fractional slot concentrated two layer winding, the fundamental winding factor is given by:

$$\xi = \xi_p \cdot \xi_d = \sin\left(\frac{N_p \pi}{Q_s}\right) \cdot \frac{\sin\left(\frac{\pi}{6}\right)}{z \cdot \sin\left(\frac{\pi}{6 \cdot z}\right)} \tag{2}$$

where $z$ is the numerator of $q$ reduced to the lowest terms:

$$q = \frac{Q_s}{6 \cdot N_p} = \frac{z}{n}. \tag{3}$$

Table 2 gives the number of poles that result in a fundamental winding factor higher than 0.866 for a stator with 24 slots.

**Table 2.** Number of poles with $\xi > 0.866$ for a stator with 24 slots.

| $Q_s$ | | **Number of Poles (p)** | | | | | | | |
| --- | --- | --- | --- | --- | --- | --- | --- | --- | --- |
| | | **16** | **18** | **20** | **22** | **26** | **28** | **30** | **32** |
| 24 | $\xi$ | 0.866 | | 0.933 | 0.9495 | 0.9495 | 0.933 | | 0.866 |
| | $k'_\phi$ | 0.324 | | 0.353 | 0.358 | 0.354 | 0.346 | | 0.329 |
| | $k_\phi = k'_\phi \cdot \xi$ | 0.281 | | 0.329 | 0.340 | 0.336 | 0.323 | | 0.285 |

The second way is via the back-emf constant $k'_\phi$:

$$k'_\phi = \frac{E_p}{\xi \cdot \Omega},\tag{4}$$

with $E_p$ the no-load back-emf (RMS value) of a phase at the mechanical speed $\Omega$. The back-emf constant for different slot/combinations is determined using the analytical model for the flux density distribution in the iron core from [16,17], the results can be found in Table 2. The prototype YASA AFPMSM has 24 slots, 26 poles and a fundamental winding factor equal to 0.9495. Although 22 and 26 poles result in the highest fundamental winding factor, other pole pair numbers will also be considered since these have a different winding diagram. The winding diagram indicates which coils belong to each phase. A different slot/pole combination results in a different winding diagram and thus a different number of adjacent coils belonging to the same phase. It can be expected that the distance between the centre of a phase with high losses and a phase with lower losses, respectively, will influence the tangential heat transfer. This is the third way via which the slot/pole combination influences the heat transfer in the tangential direction. Figure 3 gives the winding diagrams and the number of adjacent coils belonging to the same phase $n$ for the feasible slot/pole combinations from Table 2.

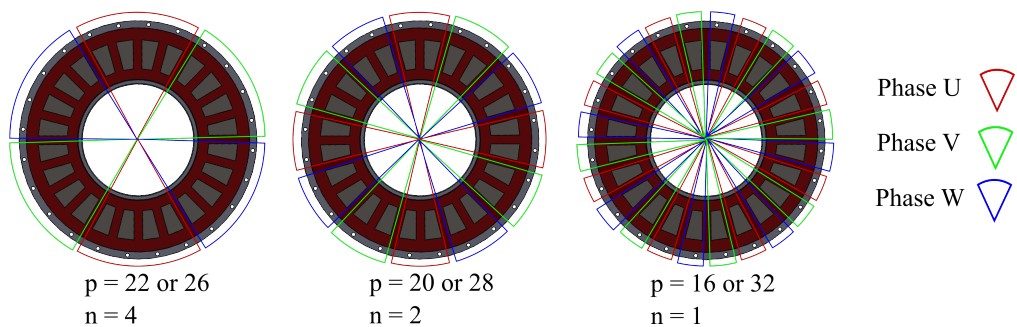

| | | |
| --- | --- | --- |
| p = 22 or 26 | p = 20 or 28 | p = 16 or 32 |
| n = 4 | n = 2 | n = 1 |

**Figure 3.** Winding diagrams and the number of adjacent coils belonging to the same phase $n$ for the feasible slot/pole combinations from Table 2.

### 2.3. Thermal End-Winding Interconnection

In Figure 4 the thermal end-winding interconnection ring (1) is shown. The concept is illustrated here for an epoxy potted YASA AFPMSM stator which is cooled via its housing at the outer diameter (3). The end-winding interconnection ring is therefore located at the inner diameter of the stator. In order to act as a highway for redistributing the heat between phases, it consists of a good thermally conducting material, e.g., aluminium or aluminium-oxide.

In [18], a copper foam end-winding interconnection ring filled with phase change material was proposed for an outer-rotor radial flux machine to improve its peak load capabilities. Different from the proposed end-winding ring in this work, the ring in [18] interconnects the end-windings of the outer-rotor radial flux machine at the coldest axial face of the motor, i.e., the face where most heat leaves the motor. Therefore, the ring only acts as a thermal buffer during peak loads, but it is less effective in redistributing the heat between phases since it is located at the colder axial side of the motor. Although the concept

of a thermal end-winding interconnection ring for redistributing the heat is illustrated in this work for a YASA AFPMSM, it is also applicable for radial flux machines.

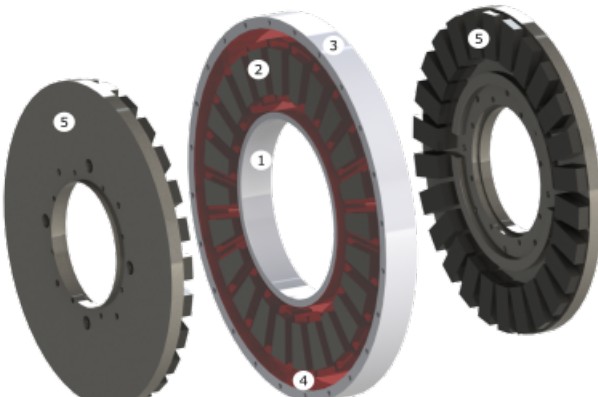

**Figure 4.** Yokeless and Segmented Armature Axial Flux Permanent Magnet Synchronous Machine: (1) Thermal end-winding interconnection ring (2) Concentrated winding tooth coil (3) Aluminium housing (4) Epoxy impregnation (5) Permanent magnet rotor.

At standstill no eddy-currents are induced in the end-winding ring; however, if the application both requires standstill and high speed operation, eddy current losses can occur in the end-winding ring if the electrical frequency is sufficiently high. The eddy current losses can be limited by choosing aluminium-oxide, or copper or aluminium foam instead of a solid aluminium ring. Therefore, this aspect will not be studied in further detail in this work.

### 2.4. Equivalent Winding Body Thermal Conductivity

A winding body consists of different materials with different thermal properties, this is often represented by equivalent thermal conductivities that depend on the thermal conductivities and volume fractions of the constituting materials [19]. Additionally, a winding body has anisotropic thermal properties due to its stranded or layered nature. Therefore, two different equivalent thermal conductivities are defined: one for the direction along the conductor and one for the direction perpendicular to the conductor. Typically, the latter is the lowest one. In previous works, it was shown that the equivalent thermal conductivity perpendicular to the conductor can be much larger for a winding body consisting of anodised aluminium [4,19]. Since the thermal conductivity of a winding body can have large influence on the thermal properties of the stator [2], it is expected that this can also have an influence on the tangential heat transfer. Therefore, the effect of the equivalent thermal conductivity of a winding body on the heat transfer in the tangential direction, and thus the stall torque performance, will be studied in this work. To this end, a YASA AFPMSM stator consisting of conventional concentrated winding tooth coils with round enamelled copper wire will be compared to a geometrically identical stator with anodised aluminium foil, which has superior equivalent thermal conductivities in both directions [4].

## 3. Materials and Methods

### 3.1. Experimental Setup

To study the influence of the slot/pole combination, the thermal end-winding interconnection and the winding body equivalent thermal conductivity on the heat transfer in the tangential direction and eventually on the stall torque performance, the experimental setup shown in Figure 5 is used. It consists only of the stator of the prototype YASA AFPMSM motor from Figure 2. The rotor is not considered in this work since at standstill the rotor losses and convective heat transfer between stator and rotor is negligible [20]. Moreover, this allows to study different slot/pole combinations with a single stator. This avoids the need to manufacture multiple rotors and/or stators and eliminates the vari-

ability in the measurements caused by variability in the rotor/stator properties due to manufacturing imperfections.

Since it was assumed that the convective heat transfer between rotor and stator is negligible, both the top and bottom airgap surface of the stator are insulated with thermal insulation wool and XPS insulation (see (4) and (5) on Figure 5). All heat leaves the stator at the outer diameter via forced convection over the housing surface. The airflow is generated by two cooling fans (see (3) on Figure 5) located above the stator.

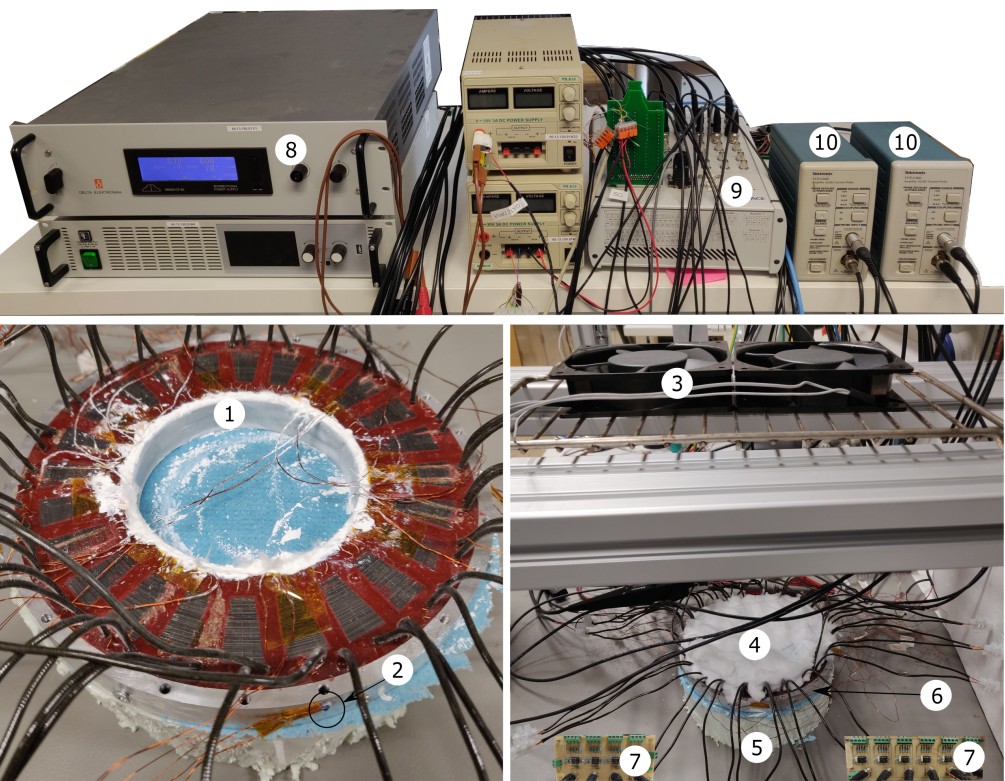

**Figure 5.** Experimental test setup. (left) Detailed view of prototype stator (1) thermal end-winding interconnection (2) PT100 temperature sensor on aluminium housing (right) Overview of setup: (3) cooling fans (4) thermal insulation wool (5) Expanded Polystyrene Insulation (XPS) (6) prototype stator between insulation (7) PT100 temperature sensor signal conditioning board (8) DC power supply (9) dSPACE MicroLabBox® real time control and data processing unit (10) Tektronix® TCPA 300 current amplifier.

As already mentioned in Section 2.1, only the worst-case uneven loss distribution has to be considered. This is the position where $I_u = 2 \cdot I_v = 2 \cdot I_w = \sqrt{2} \cdot I_{nom,RMS}$ [6]. To emulate this worst-case situation, phase $U$ is connected in series with a parallel connection of phases $V$ and $W$, as shown in Figure 6. All coils belonging to the same phase are connected in series. A programmable DC power supply is used to inject a constant current in phase $U$. This way, the same loss distribution is obtained, as if the motor would produce a certain torque at standstill.

To study the influence of the equivalent winding body thermal conductivity, two geometrically identical stators were constructed: one with enamelled round copper wire as conductor and one with anodised aluminium foil as conductor. Figure 7 shows a single tooth coil for both conductor types and Table 3 provides the detailed specifications for each tooth coil. Note that both tooth coils have the same number of turns and approximately the same resistance. This means both coils produce approximately the same losses for a given current (and thus torque). The main reason for this is the larger cross-section of the aluminium foil conductor, which is possible due to a more space-efficient stacking of the foil and a thinner electrical insulation layer. For mechanical stability reasons both stators

were impregnated in an epoxy resin; therefore, they exhibit the same appearance after impregnation and only one stator is shown in Figure 5.

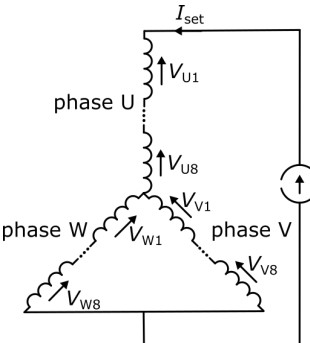

**Figure 6.** Phase connection to emulate worst-case loss distribution.

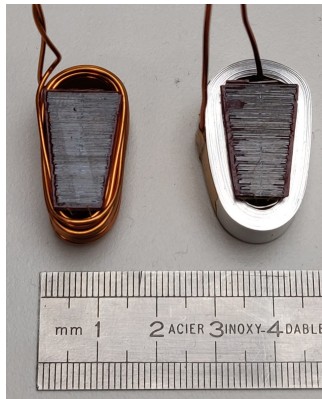

**Figure 7.** Prototype tooth coils. Left: enamelled copper wire tooth coil, Right: anodised aluminium foil tooth coil.

Since both terminals of each coil are accessible, all slot/pole combinations of Figure 3 can be realised experimentally. It also enables the measurement of the voltage over every single tooth coil and thus the determination of the dissipated power in each tooth coil. This enables the consideration of the difference in resistance between tooth coils due to a different average coil temperature.

To study the influence of a thermal end-winding interconnection, a removable aluminium ring with a thickness of 3.5 mm is used at the inner diameter of the stator (see (1) on Figure 5). Thermal paste is used to ensure a good thermal contact between the ring and the end-winding region. Since the stator with round enamelled copper wire and the stator with anodised aluminium foil are geometrically identical, the ring fits in both stators.

Every single tooth coil of both stator variants is instrumented with a PT100 temperature sensor at the inner diameter of each iron core segment (see Figure 8). This is the hotspot location in a stator in case of uniform losses [2]. Additionally, PT100 temperature sensors have been placed at certain locations at the outer diameter of the aluminium housing as well (e.g., see (2) in Figure 5). Due to spatial periodicity in the worst-case loss distribution, it is not necessary to measure the temperature of every tooth coil. For the slot/pole combinations that result in 4 and 2 adjacent tooth coils ($n = 4$ or 2), it is sufficient to consider only one quarter of the stator. For $n = 1$, 1/8th of a stator is even sufficient. To account for variations in the thermal properties between coils due to manufacturing imperfections, the hotspot temperature (i.e., the temperature in the centre of phase $U$) in two different spatial periods of the stator is measured. At the same angular location as the hotspot, also the temperature at the outer diameter of the housing is measured.

**Table 3.** Specifications of the prototype tooth coil.

| Enamelled Copper Wire (Grade I, IEC 60317-13) | Symbol | Value | Unit |
|---|---|---|---|
| Number of turns | $n_{\text{turns}}$ | 35 | / |
| Nominal outer diameter | $d_{\text{Cu,o}}$ | 0.8425 | mm |
| Conductor diameter | $d_{\text{Cu,i}}$ | 0.8 | mm |
| Winding length (incl. terminals) | $l_{\text{Cu}}$ | 276 | cm |
| measured DC resistance (@ 25 °C) | $R_{\text{DC,Cu}}$ | 94.54 ± 0.37 [1] | mΩ |
| Height laminated iron core | $h_{\text{core}}$ | 20 | mm |
| Weight of tooth coil | $m_{\text{Cu+SiFe}}$ | 31.3 | g |
| Resistivity copper | $\rho_{\text{Cu}}$ | $1.72 \times 10^{-8}$ | Ωm |
| Resistance temperature coeff. | $\alpha_{\text{Cu}}$ | $3.93 \times 10^{-3}$ | K$^{-1}$ |
| Fill factor | $f_{\text{Cu,coil}}$ | 49 | % |
| Dielectrical strength (IEC 60317-0-1) | $E_{\text{max}}$ | 87 | $V_{\text{RMS}}/\mu m$ |
| Price/kg | | 16.64 | EUR/kg |
| **Anodised aluminium foil** | | | |
| Number of turns | $n_{\text{turns}}$ | 35 | / |
| foil width | $h_{\text{Al}}$ | 10 | mm |
| total foil thickness | $t_{\text{Al,tot}}$ | 86 | m |
| thickness Al$_2$O$_3$ layer | $t_{\text{AlOx}}$ | 4.6 | m |
| Foil length (excl. terminals) | $l_{\text{Cu}}$ | 250 | cm |
| Cu terminal length (dia. 0.9 mm) | $l_{\text{term}}$ | 40 | cm |
| measured DC resistance (@ 25 °C) | $R_{\text{DC,Al}}$ | 95.83 ± 0.6 [1] | mΩ |
| Height laminated iron core | $h_{\text{core}}$ | 20 | mm |
| Weight of tooth coil | $m_{\text{Al+SiFe}}$ | 25.8 | g |
| Resistivity aluminium | $\rho_{\text{Al}}$ | $2.74 \times 10^{-8}$ | Ωm |
| Resistance temperature coeff. | $\alpha_{\text{Al}}$ | $4.03 \times 10^{-3}$ | K$^{-1}$ |
| Fill factor | $f_{\text{Al,coil}}$ | 75 | % |
| Dielectrical strength (ISO 2376) | $E_{\text{max}}$ | 26.5 | $V_{\text{RMS}}/\mu m$ |
| Price/kg | | 685 [2] | EUR/kg |

[1] mean and standard deviation over 24 tooth coils. [2] Note that this is the cost for a small order quantity of 2 kg; for larger order quantities, the cost will be lower.

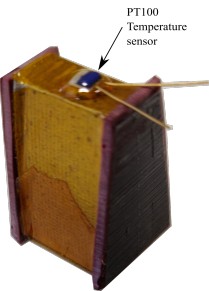

**Figure 8.** Indication of the PT100 temperature sensor location.

Besides temperature measurements, also the voltage over every single tooth coil in the half stator of Figure 5 is measured. The currents in phase *U* and *V* are measured using a Tektronix® TCPA 300 current amplifier. A dSPACE MicroLabBox® platform was used for data-acquisition. All signals were sampled at 1 kHz.

### 3.2. 3D Thermal FE Model

To support the analysis of the experimental data, a 3D thermal FE model is developed in this section. To support the experimental data analysis, the model will first be calibrated using experimental data to find the model parameters that result in a good agreement between model predictions and measurements.

### 3.2.1. Geometry

As already mentioned in Section 3.1, it is sufficient to consider only one quarter of a full stator due to the periodicity in the worst-case loss distribution. The full geometry of the FE model thus consists of 6 adjacent tooth coils. Figure 9 depicts a single tooth coil of the modelled geometry of the 3D thermal FE model. It consists of a winding body of either round enamelled copper wire or anodised aluminium foil (5) wound on a laminated iron core (2). A mica sheet (3) with a thickness of 0.2 mm is used as inter coil insulation and a solid aluminium-oxide pad (6) is used as electrical insulation between coil and housing (phase-to-ground insulation). Optionally, a thermal end-winding interconnection ring (1) can be included in the model as well. The coil assembly was potted in a low viscosity epoxy resin.

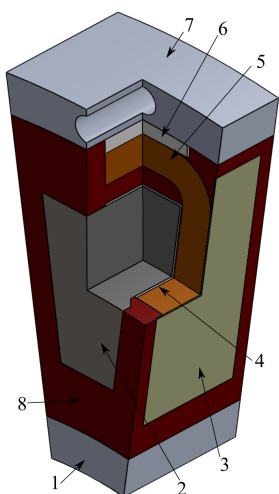

**Figure 9.** Concentrated winding tooth coil: (1) Aluminium thermal end-winding interconnection ring; (2) Laminated iron core; (3) Mica inter coil insulation sheet; (4) Mica slot liner; (5) Winding body; (6) Aluminium-oxide thermal pad; (7) Aluminium housing; (8) Epoxy potting.

### 3.2.2. Thermal Interfaces

The gap between the end-winding and the thermal end-winding ring is filled with epoxy. This gap is modelled in full detail and can be varied from 0 to 3.7 mm.

Since the phase-to-ground insulation consists of a rigid, non-deformable material (aluminium-oxide), the low viscosity potting resin also flows in the small gaps between the winding body and the housing. Since it is not feasible to model these gaps in full detail, a fill factor $f_{pad}$ is introduced:

$$f_{pad} = \frac{t_{AlOx\text{-}pad}}{t_{AlOx\text{-}pad} + t_{gap,equi}}, \tag{5}$$

with $t_{AlOx\text{-}pad}$ as the thickness of the aluminium-oxide pad between winding and housing and $t_{gap,equi}$ the equivalent thickness of the gaps filled with epoxy. The phase-to-ground insulation will be modelled as an amalgam of aluminium-oxide and epoxy resin. The fill factor will be used for the calculation of the equivalent thermal conductivity in Section 3.2.3.

### 3.2.3. Anisotropic Material Modelling

The winding body, the laminated iron core and the phase-to-ground insulation exhibit anisotropic thermal properties since these bodies consist of strands or laminations of different materials with different thermal properties. As already explained in Section 2.4, this will be modelled via two equivalent thermal conductivities. One for the direction along the strand or lamination $k_1$ and one for the direction perpendicular to the strand or out of the lamination plane $k_2$.

For the epoxy impregnated, enamelled copper wire winding body, the equivalent thermal conductivities are calculated through the use of the Hashin and Shriktman approximation [19]:

$$k_{1,\text{wi}}^{\text{Cu}} = f_{\text{wi}}^{\text{Cu}} \cdot k_{\text{Cu}} + (1 - f_{\text{wi}}^{\text{Cu}}) \cdot k_{\text{Ep}} \tag{6}$$

$$k_{2,\text{wi}}^{\text{Cu}} = k_{\text{Ep}} \cdot \frac{(1 + f_{\text{wi}}^{\text{Cu}}) \cdot k_{\text{Cu}} + (1 - f_{\text{wi}}^{\text{Cu}}) \cdot k_{\text{Ep}}}{(1 - f_{\text{wi}}^{\text{Cu}}) \cdot k_{\text{Cu}} + (1 + f_{\text{wi}}^{\text{Cu}}) \cdot k_{\text{Ep}}} \tag{7}$$

For the anodised aluminium foil winding body, a two-step approach is followed: first, the equivalent properties of the anodised foil are calculated and subsequently, these properties are used to calculate the properties of an impregnated anodised aluminium foil winding body.

Step 1:

$$k_{1,\text{afol}} = f_{\text{afol}} \cdot k_{\text{Al}} + (1 - f_{\text{afol}}) \cdot k_{\text{AlOx,film}} \tag{8}$$

$$k_{2,\text{afol}} = \frac{k_{\text{Al}} \cdot k_{\text{AlOx,film}}}{(1 - f_{\text{afol}}) \cdot k_{\text{Al}} + f_{\text{afol}} \cdot k_{\text{AlOx,film}}}, \tag{9}$$

where $k_{\text{Al}}$ and $k_{\text{AlOx}}$ are the thermal conductivity of aluminium and aluminium-oxide film, respectively. The fill factor for this combination is defined as:

$$f_{\text{afol}} = \frac{t_{\text{Al,tot}} - 2 \cdot t_{\text{AlOx}}}{t_{\text{Al,tot}}}, \tag{10}$$

where $t_{\text{Al,tot}}$ is the thickness of the anodised aluminium foil and $t_{\text{AlOx}}$ the thickness of the aluminium-oxide insulation film on the foil.

Step 2:

$$k_{1,\text{wi}}^{\text{Al}} = f_{\text{wi}}^{\text{Al}} \cdot k_{1,\text{afol}} + (1 - f_{\text{wi}}^{\text{Al}}) \cdot k_{\text{Ep}}, \tag{11}$$

where $f_{\text{wi}}^{\text{Al}}$ is the fill factor of the impregnated winding body, $k_{1,\text{afol}}$, the equivalent thermal conductivity of the anodised foil in the plane of the foil and $k_{\text{Ep}}$, the thermal conductivity of the epoxy resin. For the thermally poor conducting direction, the equivalent thermal conductivity can be calculated using the series material model from [19]:

$$k_{2,\text{wi}}^{\text{Al}} = \frac{k_{2,\text{afol}} \cdot k_{\text{Ep}}}{(1 - f_{\text{wi}}^{\text{Al}}) \cdot k_{2,\text{afol}} + f_{\text{wi}}^{\text{Al}} \cdot k_{\text{Ep}}}, \tag{12}$$

where $k_{2,\text{afol}}$ is the equivalent thermal conductivity of the anodised foil perpendicular to the plane of the foil.

The laminated iron core is an amalgam of electrical steel sheets and a thin adhesive layer to bond the sheets together. The equivalent thermal conductivities are also calculated using the series-parallel model from [19]:

$$k_{1,\text{core}} = f_{\text{core}} \cdot k_{\text{FeSi}} + (1 - f_{\text{core}}) \cdot k_{\text{Ep}} \tag{13}$$

$$k_{2,\text{core}} = \frac{k_{\text{FeSi}} \cdot k_{\text{Ep}}}{(1 - f_{\text{core}}) \cdot k_{\text{FeSi}} + f_{\text{core}} \cdot k_{\text{Ep}}}, \tag{14}$$

where $f_{\text{core}}$ is the fill factor of the laminated iron core and $k_{\text{FeSi}}$ the thermal conductivity of electrical steel.

As already mentioned in Section 3.2.2, the phase-to-ground insulation will be modelled as an amalgam of aluminium-oxide and epoxy resin. The equivalent thermal conductivities are again calculated using the series-parallel model from [19]:

$$k_{1,\text{pad}} = f_{\text{pad}} \cdot k_{\text{AlOx}} + (1 - f_{\text{pad}}) \cdot k_{\text{Ep}} \tag{15}$$

$$k_{2,\text{pad}} = \frac{k_{\text{Ep}} \cdot k_{\text{AlOx}}}{(1 - f_{\text{pad}}) \cdot k_{\text{AlOx}} + f_{\text{pad}} \cdot k_{\text{Ep}}}. \tag{16}$$

### 3.2.4. Boundary Conditions

As already explained in Section 3.1, convective heat transfer at the airgap surface is neglected. It is assumed that all heat is dissipated via forced convection over the housing at the outer diameter.

### 3.2.5. Transient Model Calibration

To calibrate the 3D thermal FE model, the coils of both the stator with round enamelled copper wire and the stator with anodised aluminium are configured to emulate a 24 slot, 22 or 26 poles combination as depicted in Figure 10. Initially, the stator is in thermal equilibrium with its environment. A constant current $I_{\text{set}} = 4A$ is then injected in phase $U$ which corresponds to 21 W dissipated power in the motor. For 100 min, the hotspot and corresponding housing temperatures of selected coils, all coil voltages, and the phase currents are recorded. The voltage and current measurements are used to calculate the power losses at every time step, these serve as inputs for the FE model transient simulation. These transient experiments are performed once for a stator without thermal end-winding interconnection ring and once for a stator with thermal end-winding interconnection ring. The results from both the simulation with the 3D thermal FE model and the measurements are given in Figures 11 and 12. The numbering of the temperature signals corresponds to the numbering of the tooth coils in Figure 10. The parameters used for the simulations are given in Table 4. Due to the temperature limit of the used epoxy resin, the maximum hotspot temperature was kept below 80 °C. It can be concluded that the simulations are in good agreement with the measurements for both the round enamelled copper wire stator and the anodised aluminium foil winding stator, and for the case with and without thermal end-winding interconnection ring. The 3D thermal FE model can thus reliably be used to support the experimental data analysis in order to obtain a better understanding of the measurement results.

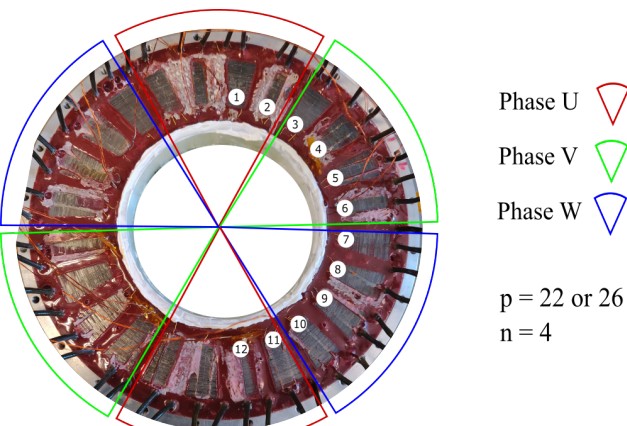

**Figure 10.** Indication of the PT100 temperature sensor location.

**Table 4.** Thermal Finite Element Model Parameters.

| Parameter | Value | Unit |
|:---:|:---:|:---:|
| $k_{\mathrm{Cu}}$ | 385 | W/mK |
| $k_{\mathrm{Al}}$ | 237 | W/mK |
| $k_{\mathrm{Ep}}$ | 0.37 | W/mK |
| $k_{\mathrm{AlOx,film}}$ | 1.6 | W/mK |
| $k_{\mathrm{AlOx}}$ | 20 | W/mK |
| $k_{\mathrm{FeSi}}$ | 28 | W/mK |
| $f_{\mathrm{wi}}^{\mathrm{Cu}}$ | 0.49 | [/] |
| $f_{\mathrm{afol}}$ | 0.89 | [/] |
| $f_{\mathrm{wi}}^{\mathrm{Al}}$ | 0.75 | [/] |
| $f_{\mathrm{core}}$ | 0.98 | [/] |
| $f_{\mathrm{pad}}$ | 0.88 | [/] |
| $k_{1,\mathrm{wi}}^{\mathrm{Cu}}$ | 189 | W/mK |
| $k_{2,\mathrm{wi}}^{\mathrm{Cu}}$ | 1.08 | W/mK |
| $k_{1,\mathrm{afol}}$ | 212 | W/mK |
| $k_{2,\mathrm{afol}}$ | 14.3 | W/mK |
| $k_{1,\mathrm{wi}}^{\mathrm{Al}}$ | 159 | W/mK |
| $k_{2,\mathrm{wi}}^{\mathrm{Al}}$ | 1.37 | W/mK |
| $k_{1,\mathrm{core}}$ | 27.4 | W/mK |
| $k_{2,\mathrm{core}}$ | 0.37 | W/mK |
| $k_{1,\mathrm{pad}}$ | 17.6 | W/mK |
| $k_{2,\mathrm{pad}}$ | 2.72 | W/mK |

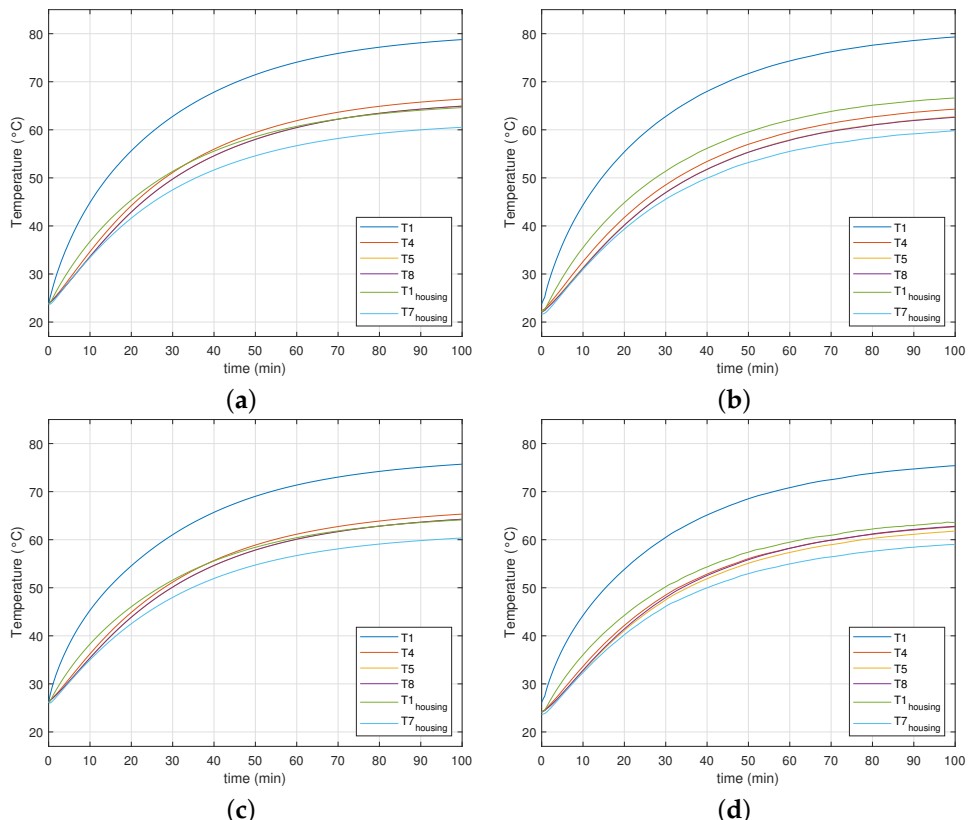

**Figure 11.** Transient temperature evolution at selected locations (see Figure 10) in a stator with round enamelled copper wire for a 24 slot, 22 or 26 pole combination. (**a**) Simulated temperatures | stator without end-winding ring. (**b**) Measured temperatures | stator without end-winding ring. (**c**) Simulated temperature | stator with end-winding ring. (**d**) Measured temperatures | stator with end-winding ring.

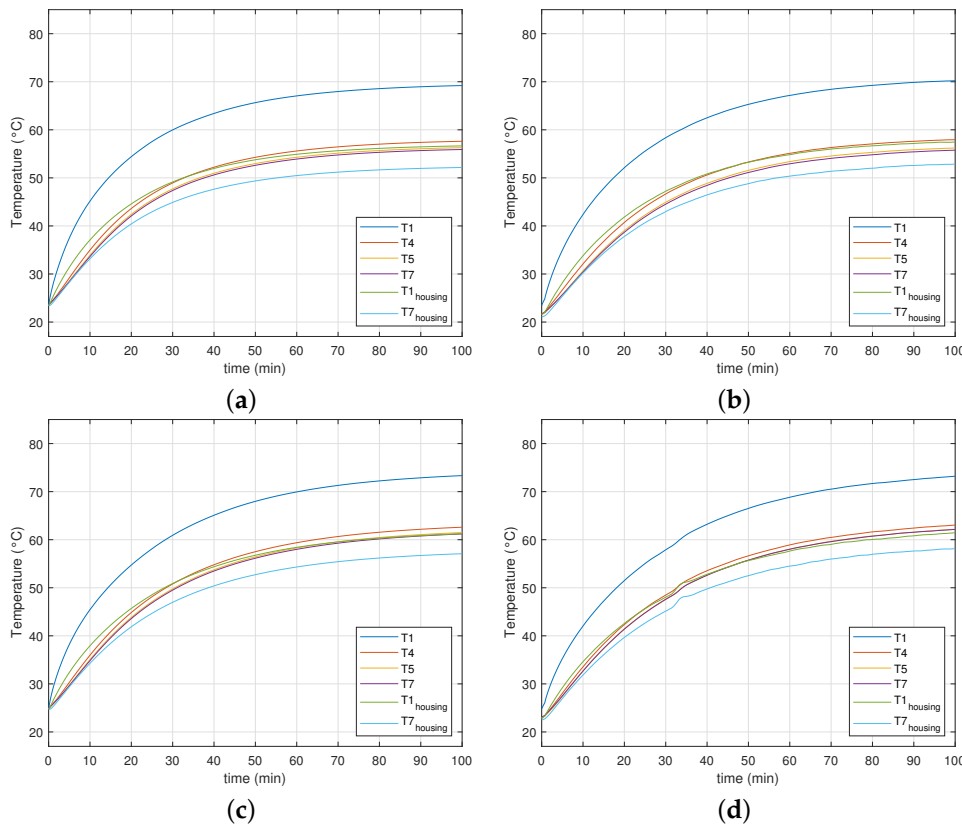

**Figure 12.** Transient temperature evolution at selected locations (see Figure 10) in a stator with anodised aluminium foil winding for a 24 slot, 22 or 26 pole combination. (**a**) Simulated temperatures | stator without end-winding ring. (**b**) Measured temperatures | stator without end-winding ring. (**c**) Simulated temperatures | stator with end-winding ring. (**d**) Measured temperatures | stator with end-winding ring.

### 3.2.6. Steady-State Temperature Distribution

Using the calibrated model from Section 3.2.5, the steady-state temperature distribution can be calculated. This is illustrated for the scenario from Figure 11a, i.e., the stator with round enamelled copper wire without thermal end-winding interconnection, with a 24 slots, 22 or 26 poles combination in Figure 13.

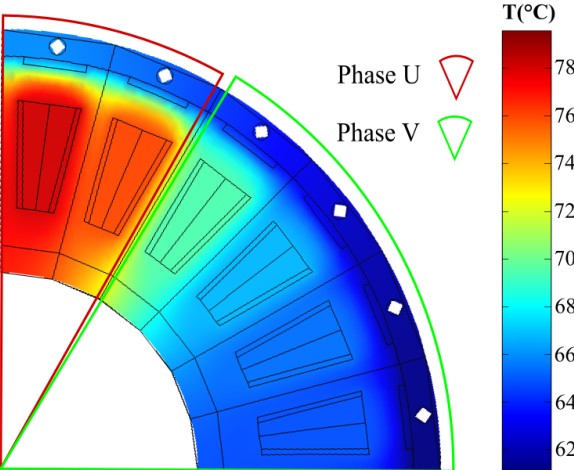

**Figure 13.** Steady-state temperature distribution in the stator with round enamelled copper wire without thermal end-winding interconnection, with a 24 slots, 22 or 26 poles combination.

As expected, the highest temperatures are found in the coils belonging to phase $U$ since the losses in phase $U$ are four times higher than in phase $V$. It can also be observed that there is a temperature difference between the coils of phase $U$ clearly proving the presence of a heat flux in the tangential direction, i.e., heat is redistributed from coils with higher losses to coils with lower losses.

## 4. Results

### 4.1. Experimental Results

To study the influence of the slot/pole combination, thermal end-winding interconnection and equivalent winding body thermal conductivity on the stall torque performance, 10 experiments are performed in which the slot/pole combination, the presence of a thermal end-winding ring and the conductor material are varied. Table 5 specifies the test conditions for each experiment.

**Table 5.** Measured thermal resistance between hotspot and housing surface.

| $n$ | Thermal End-Winding Connection (Yes/No) | Al/Cu | $R_{\text{hotspot}}$ (K/W) | $\frac{Q_{\text{stall}}}{2 \cdot Q_{\text{uniform,s24p22}}}$ |
|---|---|---|---|---|
| 4 | No | Cu | 7.89 | 0.534 |
| 4 | Yes | Cu | 7.29 | 0.578 |
| 2 | No | Cu | 6.44 | 0.655 |
| 2 | Yes | Cu | 6.34 | 0.665 |
| 1 | No | Cu | 4.86 | 0.866 |
| 1 | Yes | Cu | 4.90 | 0.860 |
| 4 | No | Al | 7.33 | 0.536 |
| 4 | Yes | Al | 6.43 | 0.611 |
| | uniform losses | Cu | 8.42 | |
| | uniform losses | Al | 7.86 | |

To emulate a stall torque loss distribution, the stator coils are connected as shown in Figure 6. A constant current $I_{\text{set}} = 4A$ is injected in phase $U$, which corresponds to 21 W dissipated power in the motor. The stall torque temperature distribution will be compared with the temperature distribution under uniform losses. To emulate this scenario, all coils of the stator are connected in series and a constant current $I_{\text{set}}^{\text{uniform}} = 2.8A = 4/\sqrt{2}A$ is injected into this ring network, also resulting in 21 W dissipated power. After 100 minutes, thermal steady-state is reached and the hotspot and corresponding housing temperatures, coil voltages and phase currents are recorded. The recorded values are used to calculate the thermal resistance between the hotspot and the housing at the same angular position $(\theta^*)$ as:

$$R_{\text{hotspot}} = \frac{T_{\text{hotspot}}(\theta^*) - T_{\text{housing}}(\theta^*)}{Q_{\text{coil},U}}$$

$$Q_{\text{coil},U} = V_{\text{coil},U} \cdot I_U,$$

(17)

where $T_{\text{hotspot}}(\theta^*)$ is the temperature in the hotspot of a coil (see Figure 8) in the centre of phase $U$, $T_{\text{housing}}(\theta^*)$ is the temperature of the housing (see (2) in Figure 5) at the same angular location $\theta^*$, $Q_{\text{coil},U}$ is the dissipated power in this coil, which is the product of the voltage $V_{\text{coil},U}$ over and the current $I_U$ through the coil. The thermal resistance is preferred as a metric to compare the experimental results over the absolute or relative temperature since this metric is less sensitive to variations in ambient conditions (e.g. ambient temperature and housing convective heat transfer coefficient). As already mentioned in Section 3.1, two hotspot and housing temperatures are measured in different spatial periods of the stator, the average of thermal resistance of both is given in Table 5. Note that the thermal resistance in all cases is lower compared to the thermal resistance in case of uniform losses. This could be expected since a part of the losses of phase $U$ are dissipated via the other phases, hence the decrease in the effective thermal resistance of phase $U$. Although the thermal resistance is lower in case of uneven loss distribution, the temperature is still higher

in comparison to a uniform loss distribution because two times more losses are dissipated in phase $U$ in case of uneven loss distribution. Therefore, the maximum stall torque is lower than the maximum torque at low speed. Low speed means sufficiently high such that a uniform loss distribution can be assumed, but low enough such that iron and mechanical losses can be neglected. The ratio of the maximum stall torque over the torque at low speed will be used as a performance metric to quantify the stall torque performance of a motor. This ratio can also be interpreted as a 'torque derating factor at standstill':

$$\frac{T_{\text{stall}}}{T_{\text{uniform,ref}}}, \tag{18}$$

where $T_{\text{uniform,ref}}$ is the maximum torque under uniform losses at low speed for a reference scenario. The calculation of the torque derating factor as well as the definition of the reference scenario will be discussed in detail in the following paragraphs.

### 4.1.1. Influence of Slot/Pole Combination

To study the influence of the slot/pole combination, $T_{\text{uniform,ref}} = T_{\text{uniform,s24p22}}$ with $T_{\text{uniform,s24p22}}$ the maximum torque under uniform losses at low speed for a stator without end-winding ring and for the slot/pole combination with the highest $\xi \cdot k'_\phi$, i.e. 24 slots and 22 poles. This factor is now calculated using the measured thermal resistance from Table 5 and the fundamental winding factor and back-emf constant from Table 2:

$$\frac{T_{\text{stall}}}{T_{\text{uniform,s24p22}}} = \frac{\xi \cdot k'_\phi \cdot I_{\text{stall}}}{\xi_{\text{s24p22}} \cdot k'_{\phi,\text{s24}p22} \cdot I_{\text{uniform,s24p22}}}. \tag{19}$$

The currents can be expressed in terms of the dissipated power losses in phase $U$:

$$Q_{U,\text{stall}} = R_U \cdot (1 + \alpha \Delta T_{\text{stall}}) \cdot I^2_{U,\text{stall}} \tag{20}$$

and

$$Q_{U,\text{uniform,s24p22}} = R_U \cdot (1 + \alpha \Delta T_{\text{uniform,s24p22}}) \cdot I^2_{U,\text{uniform,s24p26}} \tag{21}$$

and since for the worst-case standstill position $I_{U,\text{stall}} = \sqrt{2} \cdot I_{\text{stall}}$, Equation (20) becomes:

$$Q_{U,\text{stall}} = R_U \cdot (1 + \alpha \Delta T_{\text{stall}}) \cdot (\sqrt{2} I_{\text{stall}})^2, \tag{22}$$

with $\alpha$, the resistance temperature coefficient, $\Delta T_{\text{stall}}$ and $\Delta T_{\text{uniform,s24p22}}$, the difference between the winding temperature and the reference temperature at which the winding resistance $R_U$ was determined. To obtain a fair comparison between the cases in Table 5, it is assumed that $\Delta T_{\text{stall}} = \Delta T_{\text{uniform,s24p22}}$ and $R_U$ is assumed the same for all slot/pole combinations of the stator with round copper wire in Table 5, Equation (19) now becomes the following:

$$\frac{T_{\text{stall}}}{T_{\text{uniform,s24p22}}} = \frac{\xi \cdot k'_\phi}{\xi_{\text{s24p22}} \cdot k'_{\phi,\text{s24}p22}} \cdot \sqrt{\frac{Q_{\text{stall}}}{2 \cdot Q_{\text{uniform,s24p22}}}} \tag{23}$$

$$\frac{Q_{\text{stall}}}{Q_{\text{uniform,s24p22}}} = \frac{\frac{\Delta T_{\text{stall}}}{R_{\text{hotspot,stall}}}}{\frac{\Delta T_{\text{uniform,s24p22}}}{R_{\text{hotspot,uniform,s24p22}}}} = \frac{R_{\text{hotspot,uniform,s24p22}}}{R_{\text{hotspot,stall}}}. \tag{24}$$

The ratio $\frac{Q_{\text{stall}}}{Q_{\text{uniform,s24p22}}}$ can now be found using the thermal resistance measurements from Table 5. The assumption of equal temperature difference between hotspot and housing in the 'stall torque case' and the 'uniform losses reference case' $\Delta T_{\text{stall}} = \Delta T_{\text{uniform,s24p22}}$ entails that the hotspot temperature in both cases is the same if the housing temperature is the same. The same hotspot temperature means that in both cases the motors operate at their thermal limit.

To study the influence of the slot/pole combination on the stall torque performance, the coils of the stator of the prototype YASA AFPMSM (Figure 5) are allocated to a phase as shown in Figure 3 and their terminals are connected according to Figure 6. Experiments are performed as described in Section 4.1. First, the ratio $Q_{\text{stall}}/Q_{\text{uniform,s24p22}}/2$ is considered. This ratio is also given in Table 5 and can be interpreted as the ratio of the total losses that can be dissipated in a motor producing torque at standstill over the losses that can be dissipated in a motor with uniform loss. To have a fair comparison, in both cases, the losses result in the same temperature difference between hotspot and housing. The results from Table 5 are visualised in a bar plot in Figure 14. It is clear that the number of adjacent coils belonging to the same phase has a large influence on the ratio $Q_{\text{stall}}/Q_{\text{uniform,s24p22}}/2$. For the case $n = 4$, which corresponds to 24 slots and either 22 or 26 poles, even 47% less power can be dissipated to obtain the same difference between hotspot and housing as compared to the case of a uniform loss distribution. The slot/pole combinations with $n = 1$ exhibit the best thermal performance because losses from phase $U$ can be very effectively dissipated via the tooth coils from phases $V$ and $W$ due to the large surface area between the phases since every tooth coil from phase $U$ has two neighbouring tooth coils from other phases.

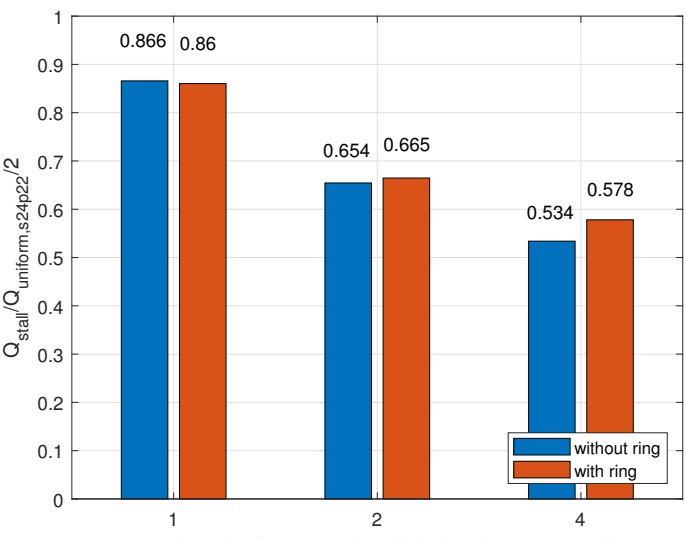

**Figure 14.** Measured ratio of the losses that can be dissipated in under stall torque conditions over the losses that can be dissipated in case of uniform loss distribution for various slot/pole combinations (see Figure 3) and for the cases with and without thermal end-winding interconnection.

The superior thermal performance of the slot/pole combinations with $n = 1$ does not necessarily imply superior stall torque performance. The torque derating factor $T_{\text{stall}}/T_{\text{uniform,s24p22}}$ for various slot/pole combinations is shown in Figure 15. It can be seen that the combination of 24 slots and 20 poles exhibits the highest stall torque performance because this combination combines good thermal performance (see Figure 14) with a good winding factor and back-emf constant. The stall torque of this slot/pole combination is 8.5%, which is higher than the slot/pole combination in the prototype YASA AFPMSM (24 slot, 26 poles). However, the differences between slot/pole combinations in terms of the stall torque performance are less pronounced than in terms of the thermal performance (see Figure 14).

### 4.1.2. Influence of Thermal End-Winding Interconnection

To study the influence of a thermal end-winding interconnection on the stall torque performance, a solid aluminium ring is inserted at the inner diameter of the stator prototypes and the same experiments as described in Sections 4.1 and 4.1.1 are repeated. The results in terms of the thermal performance were already given in Table 5 and are shown in Figure 14. The addition of a thermal end-winding ring results in 0.6% up to 4% increase

in thermal performance for $n = 1$ and $n = 4$, respectively, compared to the case without end-winding ring. This confirms that a thermal end-winding interconnection contributes in redistributing the heat from phase $U$ to the other phases. The influence is more pronounced for $n = 4$ because in this case the surface area for heat transfer between phases is limited and thus, adding a thermal end-winding interconnection has more impact on the thermal performance. However, compared to the influence of the slot/pole combination on the thermal performance or stall torque performance, the influence of an end-winding ring is lower. A possible explanation for this is the presence of a large gap (2 mm) filled with epoxy between the end-winding and the end-winding interconnection ring. The influence of this gap will be studied in more detail in Section 4.2.1.

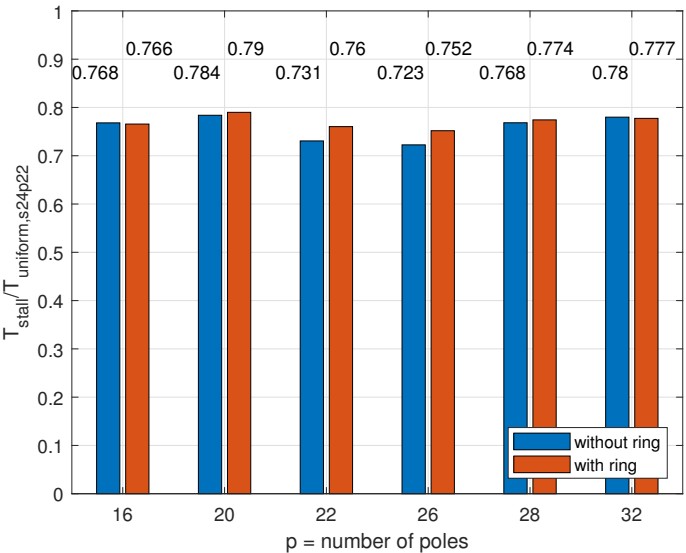

**Figure 15.** Measured torque derating factor $T_{\text{stall}}/T_{\text{uniform,s24p22}}$ for various slot/pole combinations and for the case with and without thermal end-winding interconnection.

### 4.1.3. Influence of Equivalent Winding Body Thermal Conductivity

The influence of the equivalent thermal conductivity of the winding body is studied by comparing a stator with round enamelled copper wire as conductor and a stator with anodised aluminium foil as conductor. The experiments as described in Section 4.1 are performed on both stators for $n = 4$ (see Figure 3), once without the end-winding interconnection and once with the end-winding interconnection. The outcome of the experiments was already given in Table 5. $T_{\text{uniform,ref}}$ in the torque derating factor $T_{\text{stall}}/T_{\text{uniform,ref}}$ is now defined as $T^{\text{x}}_{\text{uniform,s24p22}}$ with $x \in \{\text{Al,Cu}\}$ i.e., the maximum torque under uniform losses at low speed for either the stator with anodised aluminium foil winding or with round enamelled copper wire, without end-winding ring and the slot/pole combination with the highest $\xi \cdot k'_\phi$, i.e., 24 slots and 22 poles. The results are visualised in Figure 16.

No significant difference can be observed between the two conductor types. Although, it was expected that the stator with anodised aluminium foil winding would have a better stall torque performance due to its higher equivalent thermal conductivity in the direction out of the plane of the foil, i.e., $k^{\text{Al}}_{2,\text{wi}}$. However, since no explanation can be given at this point, this aspect will be further investigated with the aid of the 3D thermal FE model in Section 4.2.2.

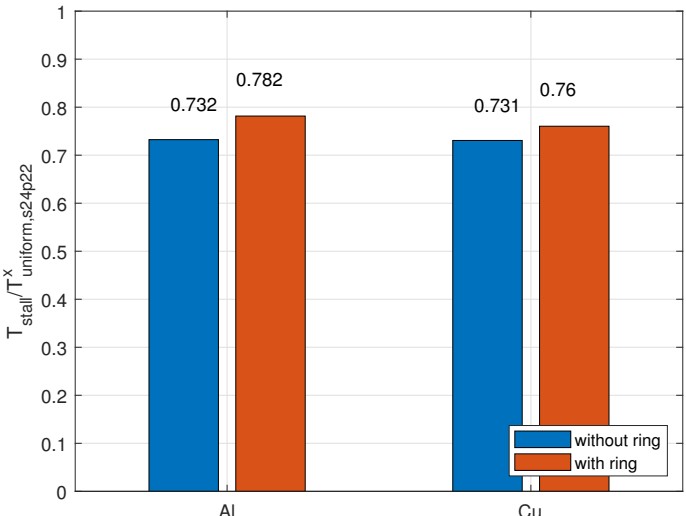

**Figure 16.** Measured torque derating factor $T_{\text{stall}}/T^{\text{x}}_{\text{uniform,s24p22}}$ with $\text{x} \in \{\text{Al,Cu}\}$ for different conductor types, for $n = 4$, and for the case with and without thermal end-winding ring.

### 4.2. Experimental Data Analysis through Simulation

4.2.1. Influence of Gap between End-Winding and End-Winding Interconnection Ring

It was mentioned in Section 4.1.2 that the addition of a thermal end-winding interconnection ring had less influence on the stall torque performance in comparison to the choice of the slot/pole combination. The presence of a gap filled with epoxy in between the end-winding and the ring was mentioned as a possible explanation. To study whether a higher stall torque performance can be obtained through the addition of a thermal end-winding ring if this gap is thinner, the calibrated 3D thermal FE model from Section 3.2.5 will be used here. The gap between the end-winding and the end-winding ring is decreased to its minimal value, the power losses in each tooth coil recorded during the experiments described in Section 4.1 are used as inputs for the simulation. The torque derating factor $T_{\text{stall}}/T_{\text{uniform,s24p22}}$ is again calculated using the simulated steady-state temperature distribution for various slot/pole combinations for the stator with copper wire winding for both the case without end-winding ring and the case with a minimal gap between the end-winding and end-winding ring. The results are given in Figure 17a.

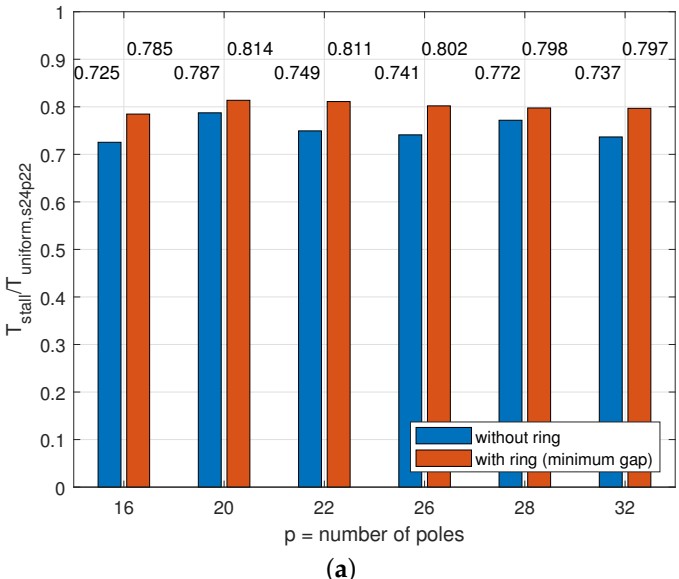

**(a)**

**Figure 17.** *Cont.*

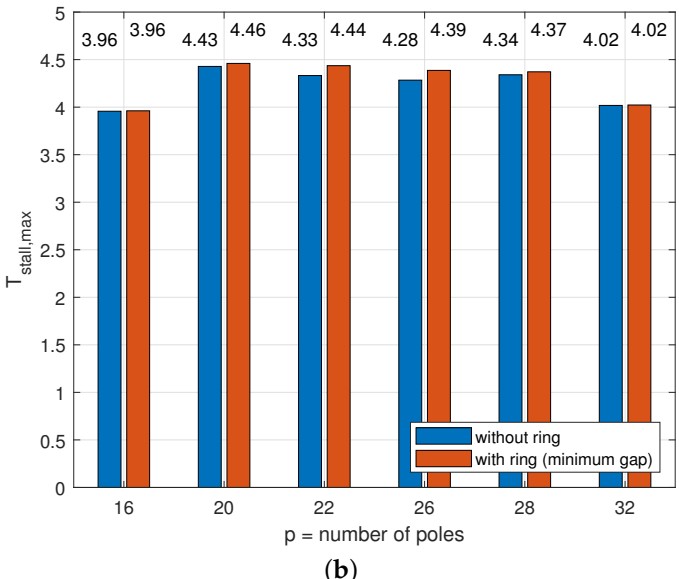

(**b**)

**Figure 17.** Simulated (**a**) torque derating factor $T_{\text{stall}}/T_{\text{uniform,s24p22}}$ and (**b**) maximum stall torque for various slot/pole combinations and for the case without thermal end-winding interconnection and with a thermal end-winding interconnection with a minimum gap between the end-winding and the ring. (**a**) Simulated torque derating factor $T_{\text{stall}}/T_{\text{uniform,s24p22}}$. (**b**) Simulated maximum stall torque (@ $T_{\text{hotspot}} = 150\,^\circ\text{C}$).

The addition of a thermal end-winding interconnection ring with minimum gap between end-winding and ring allows to increase the stall torque performance by up to 8.2%, whereas this was only 4% with a gap of two millimetres as mentioned in Section 4.1.2.

To compare the actual stall torque values for the different cases, the maximum stall torque for the various cases is shown in Figure 17b. The maximum stall torque is defined as the torque that results in a hotspot temperature of 150 °C. This torque was calculated using the current that results in a hotspot temperature of 150 °C, Equation (1) and the values from Table 2. The current was found iteratively using the thermal FE model. It can be seen that there is a difference of 0.5 Nm in maximum stall torque between the slot/pole combinations 24/16 and 24/22.

### 4.2.2. Analysis of Equivalent Thermal Conductivity of Winding Body

The study of the influence of the equivalent winding body thermal conductivity is one of the main goals of this work; however, the experimental results from Section 4.1.3 did not allow us to draw strong conclusions. No clear difference in stall torque performance between the stator with copper wire winding and the stator with anodised aluminium foil could be found. It was expected that the higher equivalent thermal conductivity of the anodised foil in the direction out of the plane of the foil [4,19] would lead to a better tangential heat transfer. This could not be proven experimentally.

By varying the winding fill factor in the 3D thermal FE model, the equivalent winding body thermal conductivity can be varied. The fill factor will be varied by ±20% with respect to its nominal value in the 3D thermal FE model and the torque derating factor $T_{\text{stall}}/T^{\text{x}}_{\text{uniform,s24p22}}$ with x ∈ {Al,Cu} is again calculated based on the simulated steady-state temperature distribution. The recorded power losses from the experiments in Section 4.1.3 were used as inputs for the simulations. The results are visualised in the barplot in Figure 18.

It can be concluded that for both the aluminium winding and copper winding stators, the fill factor and the the equivalent winding body thermal conductivity has no significant influence on the stall torque performance.

It is important to note, however, that increasing the fill factor increases both the maximum stall torque $T_{\text{stall}}$ and the maximum torque under uniform losses at low speed

$T^{x}_{\text{uniform,s24p22}}$ in the ratio $T_{\text{stall}}/T^{x}_{\text{uniform,s24p22}}$ with $x \in \{\text{Al,Cu}\}$, but it will not allow a higher maximum stall torque as compared to its maximum torque under uniform losses at low speed.

Similarly to in Section 4.2.1, the actual stall torque values for the different cases are compared in Figure 18. It can be seen that, also at elevated temperatures, the winding body thermal conductivity does not have a significant influence on the stall torque values. This is because the winding body thermal resistance is not dominant in the thermal path from heat source to heat sink. Therefore, changes in the thermal conductivity do not have a significant impact on the thermal performance and thus, has no impact on the stall torque performance either. It could already be seen in Figures 11 and 12 that the temperature difference between hotspot and housing was much smaller than the difference between hotspot and ambient, confirming that the thermal resistance between housing and ambient is dominant in this case.

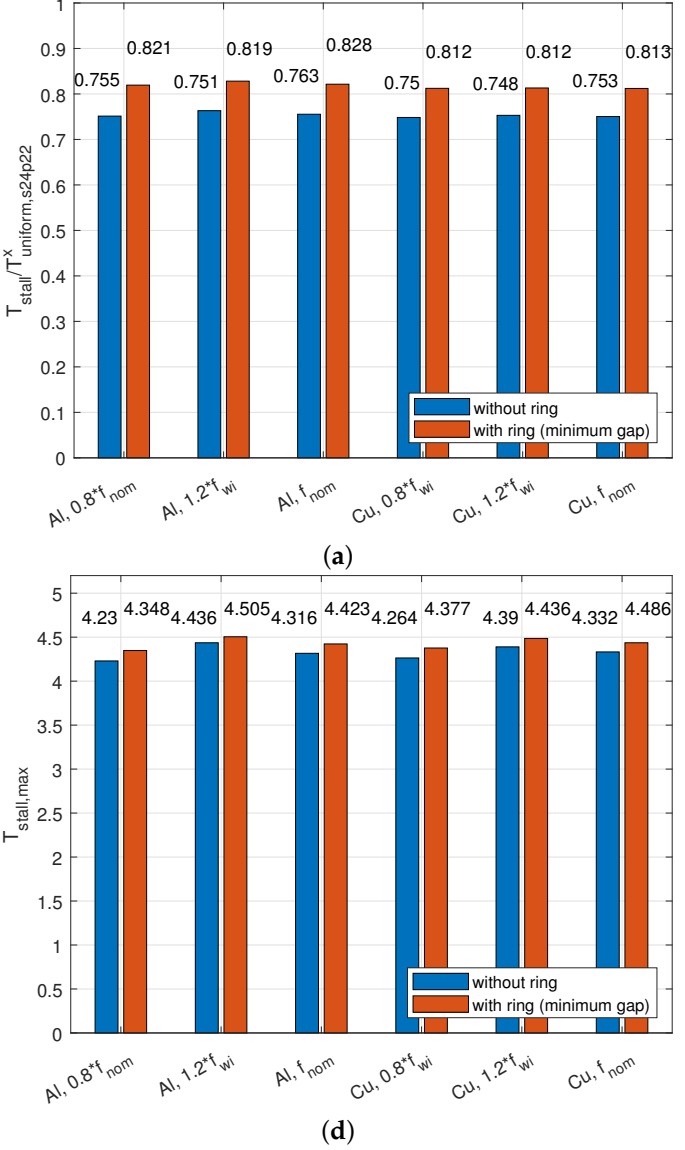

**Figure 18.** Simulated (**a**) torque derating factor $T_{\text{stall}}/T^{x}_{\text{uniform,s24p22}}$ with $x \in \{\text{Al,Cu}\}$ and (**b**) maximum stall torque, for different values of $f^{\text{Al}}_{\text{wi}}$, for different conductor types, for a 24 slots and 22 or 26 poles combination, and for the case with and without thermal end-winding ring. (**a**) Simulated torque derating factor $T_{\text{stall}}/T_{\text{uniform,s24p22}}$. (**b**) Simulated maximum stall torque (@ $T_{\text{hotspot}} = 150\,^{\circ}\text{C}$).

### 4.2.3. Influence of Cyclic Loading

There are no applications in which a motor operates sufficiently long at standstill and maximum stall torque to reach thermal steady state. More frequently, as in the case of a force-controlled robotic gripper, the load cycle of the motor consists of a long period of high torque at standstill, e.g. when the gripper is holding a soft object and short periods of low torque and high speed when the gripper is opening and closing. Since the term stall torque is frequently used in motor datasheets, it is also used as a metric here, as it is a well known and easy-to-understand term. However, it would be interesting to study the impact of a load cycle on the stall torque performance. To this end, the thermal FE model is used to simulate the following load cycle which corresponds to a realistic gripper scenario: (1) gripper grasps object at start location: high speed, gradually increasing current (duration: 4 s); (2) gripper holds object while it is moved to end position: standstill, maximum current (duration: 26 s); (3) gripper releases object at end position: high speed, gradually decreasing current (duration: 4 s); (4) gripper without object returns to start position: standstill, no current (duration: 26 s). Hence, the duty cycle for this load cycle is 50%. A scenario with uniform loss distribution as would be the case if the motor rotates when generating torque and a scenario with non-uniform loss distribution as is the case for a motor generating torque at standstill are compared. For both scenarios, the torque that results in a peak hotspot temperature of 150 °C is determined. The hotspot temperature variations for both scenarios are given in Figure 19. Larger temperature variations can be found for the case with non-uniform loss distribution since the variation of the losses is larger in case of a non-uniform loss distribution. The torque derating factor for this load cycle is then calculated as the ratio of the maximum stall torque over the maximum torque in case of uniform losses, i.e. when the losses are uniform during phase (2). The derating factor for a load cycle is compared to the derating factor for a continuous load scenario, but no difference could be observed, this allows to conclude that the stall torque derating factor for continuous load can reliably be used also in cyclic load applications.

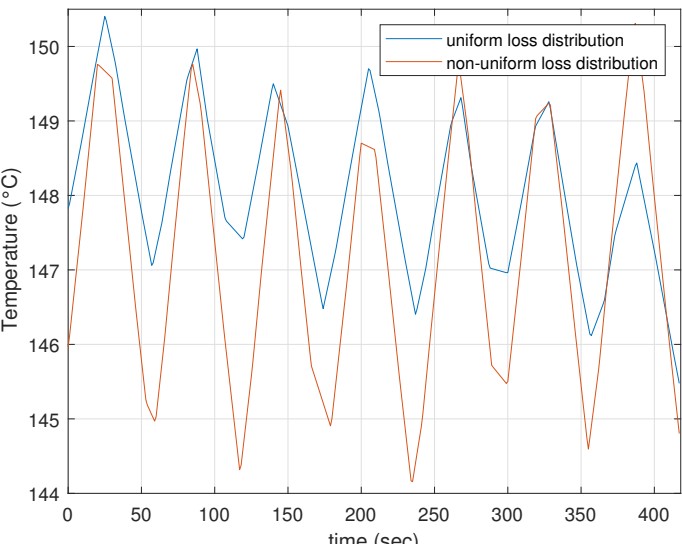

**Figure 19.** Simulated hotspot temperature variations for a load cycle with a duty cycle of 50% for a case with uniform losses (e.g., when the motor is rotating when producing torque) and for a case with non-uniform losses (e.g., when the motor is at standstill when producing torque).

## 5. Conclusions

In this work, the stall torque performance of a YASA AFPMSM motor was analysed. More specifically, the influence of the slot/pole combination, the addition of a thermal end-winding interconnection ring and the equivalent thermal conductivity of the winding body were studied. To this end, prototype YASA AFPMSM stators were manufactured and instrumented with temperature sensors. A uniform and an uneven loss distribution were

imposed on the stators by injecting DC currents to emulate a low-speed and a standstill situation, respectively. The steady-state hotspot temperatures and losses were recorded and used to calculate the thermal resistance between hotspot and housing. These thermal resistances, the fundamental winding factor and the back-emf constants were then used to calculate the stall torque performance metric introduced in this work. This metric was determined for various slot/pole combinations, a stator with/without thermal end-winding interconnection and for a stator with anodised aluminium foil winding and a stator with copper wire winding to study the impact of the equivalent thermal conductivity of the winding body.

It was concluded that only the slot/pole combination and the addition of a thermal end-winding interconnection can have a significant impact on the stall torque performance (with up to 8% increase in the stall torque performance metric). The equivalent thermal conductivity of the winding body has no impact on the stall torque performance, specifically; however, a higher fill factor leads to superior thermal properties and thus, to both a higher maximum stall torque and a higher maximum torque at low speed.

**Author Contributions:** Conceptualization, J.V.D.; methodology, J.V.D. and H.V.; software, J.V.D. and H.V.; validation, J.V.D.; formal analysis, J.V.D.; investigation, J.V.D.; resources, J.V.D.; data curation, J.V.D.; writing—original draft preparation, J.V.D.; writing—review and editing, H.V. and G.C.; visualization, J.V.D.; supervision, H.V. and G.C.; project administration, J.V.D.; funding acquisition, J.V.D., H.V. and G.C. All authors have read and agreed to the published version of the manuscript.

**Funding:** J. Van Damme was awarded a Ph.D. Fellowship Strategic Basic Research (SB) from the Research Foundation Flanders (FWO) in 2019 (Grant Number: 1S87322N). This research was also financially supported by Flanders Make vzw in the research project FiberMech.

**Data Availability Statement:** The study did not report any data.

**Acknowledgments:** The authors would like to acknowledge Tony Boone and Vincent Gevaert for their assistance in the manufacturing of the YASA AFPMSM prototypes.

**Conflicts of Interest:** The authors declare no conflict of interest.

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
