# Peer review of "Stall Torque Performance Analysis of a YASA Axial Flux Permanent Magnet Synchronous Machine"

_machines, doi:10.3390/machines11040487_

Round 1

Reviewer 1 Report

In the submission, The Yokeless and Segmented Armature Axial Flux Permanent Magnet Synchronous Machine is researched, and the results indicate that the good slot/pole combination could improve the torque performance at standstill. The comments are listed as follows

1, the introduction about the YASA AFPMSM in Figure 1 and Figure 2 should be improved, and more details about the structure should be added.

2, in the experiment part, the whole experimental setup about the YASA AFPMSM should be provided.

3, is there any comparisons to other kinds of YASA AFPMSM?

4, the influences caused by the variation of control current and rotating speed should be analyzed and considered.

Minor editing of English language required

Author Response

Dear reviewer,

Thank you for your valuable comments and suggestions to improve the work. In the attached file I have added a description of the performed changes in order to address the comments.

Yours sincerely,

Jordi Van Damme

Reviewer 2 Report

The problem titled as Stall Torque performace analysis is very complicated because it is very complex problem especially in the Synchronous PM machinhes. However the stall torque is the reault of the electric parameters of the motor windings and its current but without the speed the most critical problem is heating what may lead to the motor damage. This problem was also mentioned in the paper. 

For thermal analysis it is important to define the thermal conductivity factor what Authors done using the equvalent thermal conductivity because of the comlexity of the stator construction. This factors were used for the FE Thermal model prreparation. In my opinion description of the  proces of modeling with some data retaled to the FE model may only enrich this paper an than should be add. 

In some way the conclusions are very poor taking into account the content of the paper. As I underatnd the defined power of the losses and the calculated current were the initiol data for all thye cases. So practically the work was focused on the Investigation of the stator winding configuration influence on the stall torque relateed to defined uniform torque. It will be interested what the values had the stall tporque for analysed cases.

More over I think the Autors may try to relate the problem of the stall torque production usinf the electric energy so consuming energy with other methods like using the brakes for example the value of tis torque for both cases.

In conclusion Author did not notice the impact of the thermal conductivity on the stall torque. It may be a result of so high thermal capacity of the stator construction. Thermal balence for this systema and for the assumed current occur after 100 minuts and the temperature still was not so high. What if this temerature were higher?

Author Response

(The authors gave the same response as above.)
